# Efficient Clustering Based On A Unified View Of K-means And Ratio-cut

**Shenfei Pei**
School of Computer Science and
Center for OPTIMAL
Northwestern Polytechnical University
shenfeipei@gmail.com

**Feiping Nie**[*]
School of Computer Science and
Center for OPTIMAL
Northwestern Polytechnical University
feipingnie@gmail.com

**Rong Wang**
School of Cybersecurity and
Center for OPTIMAL
Northwestern Polytechnical University
wangrong07@tsinghua.org.cn

**Xuelong Li**
School of Computer Science and
Center for OPTIMAL
Northwestern Polytechnical University
li@nwpu.edu.cn

## Abstract

Spectral clustering and $k$-means, both as two major traditional clustering methods, are still attracting a lot of attention, although a variety of novel clustering algorithms have been proposed in recent years. Firstly, a unified framework of $k$-means and ratio-cut is revisited, and a novel and efficient clustering algorithm is then proposed based on this framework. The time and space complexity of our method are both linear with respect to the number of samples, and are independent of the number of clusters to construct, more importantly. These properties mean that it is easily scalable and applicable to large practical problems. Extensive experiments on 12 real-world benchmark and 8 facial datasets validate the advantages of the proposed algorithm compared to the state-of-the-art clustering algorithms. In particular, over 15x and 7x speed-up can be obtained with respect to $k$-means on the synthetic dataset of 1 million samples and the benchmark dataset (CelebA) of 200k samples, respectively [GitHub].

## 1   Introduction

As one of fundamental technologies in the fields of pattern recognition, machine learning, data mining, and others, clustering is getting more and more attention in recent years [29, 31]. A series of algorithms have been proposed for cluster analysis and applied to various areas successfully, such as document clustering [18], image segmentation [17], and social networks [20].

Given a set of input patterns, the purpose of clustering is to group the data into a certain number of clusters so that the samples in the same cluster are similar to each other, and the samples in different clusters are not. Among various algorithms, the classical ratio-cut and $k$-means are almost the two most popular clustering approaches.

**Ratio-cut and its related algorithms:**   Given a set of samples $\mathbf{X} = [\mathbf{x}_1, \cdots, \mathbf{x}_n]^T \in \mathbb{R}^{n \times d}$ and the number $c$ of clusters to construct, the classical ratio-cut usually consists of the following four steps:
1. Calculate the similarity matrix $\mathbf{W} \in \mathbb{R}^{n \times n}$ where $w_{ij}$ denotes the similarity between $\mathbf{x}_i$ and $\mathbf{x}_j$.
2. Compute the Laplacian matrix $\mathbf{L} = \boldsymbol{\Delta} - \mathbf{W}$, where $\boldsymbol{\Delta}$ is a diagonal matrix, and $\Delta_{ii} = \sum_{j=1}^{n} w_{ij}$.

---

[*]Corresponding author: Feiping Nie

3. Compute the first $c$ eigenvectors of its Laplacian matrix. 4. Separate these samples into $c$ classes by running $k$-means on these features. In spite of its good (promising) performance, ratio-cut and other traditional spectral clustering methods (SC) suffer from the following drawbacks: (1) The time complexity of traditional spectral clustering is $O(n^2c)$, which is one of significant drawbacks of SC. Much effort has been devoted to accelerate the process. [13] computed the approximate solution of the eigenvalue decomposition by using the classical Nyström algorithm. Based on the randomized low-rank matrix approximation method, [21] proposed a scalable Nyström scheme to further accelerate the algorithm. In addition, the anchor-based strategy [42, 41, 36] is also a very popular method for accelerating the spectral clustering method. In these methods, a similarity matrix between samples and anchors, whose size is smaller usually, to approximate the original similarity matrix. Then the clustering algorithm is performed on the reduced data. However, the performance of these methods is sensitive to how anchor-graphs are constructed and a lot of information will be lost in the sampling step. (2) To obtain the final solution, most of SC algorithms follow a two-stage approach, which may result in bad clustering structure and deviations from the solution of the original problem. Therefore [8, 48] proposed the methods where the discrete solution can be obtained directly.

$k$**-means and its related methods:**  As a well-regarded and standard approach to cluster analysis, the traditional $k$-means groups input dataset $\mathbf{X}$ into $c$ clusters by assigning each data point to the cluster with the nearest centroid. Let $\mathbf{A}_1, \cdots, \mathbf{A}_c$ denote $c$ disjoint sets, the objective function of $k$-means can be formulated as

$$\min_{\mathbf{A}_1, \mathbf{A}_2, \cdots, \mathbf{A}_c} \sum_{i=1}^{c} \sum_{\mathbf{x}_j \in \mathbf{A}_i} \|\mathbf{x}_j - \mathbf{m}_i\|_2^2, \tag{1}$$

where $\mathbf{m}_i = \sum_{\mathbf{x}_j \in \mathbf{A}_i} \mathbf{x}_j / n_i$ is the centroid of cluster $\mathbf{A}_i$ containing $n_i$ samples.

Although the problem (1) is NP-hard, many efficient optimization algorithms [24, 34, 12] are proposed, and converge quickly to a local optimum.

(1) A major disadvantage of $k$-means is that it cannot separate clusters non-linearly separable in input space. To this end, [38, 19] proposed the kernel $k$-means where a feature function $\phi : \mathbb{R}^d \to \mathcal{F}$ is adopted to map the input vectors non-linearly into feature vectors. Then traditional $k$-means is performed in the feature space instead of the input space. However, the kernel-based methods do not scale well in terms of the computational overhead. [9] proposed an approximation scheme by employing a randomized approach, to reduce the computational overhead. In addition, the Nyström and incomplete cholesky factorization methods have been adopted in [44, 16, 7] to solve kernel learning problems approximately. (2) The clustering results will be affected by the initialization of cluster centers. To this end, $k$-means++, the state-of-the-art algorithm, is proposed in [1], and its solution quality has a theoretical guarantee. Despite its strong empirical performance, $k$-means++ is difficulty scalable to large-scale problems due to its high computational complexity. A Markov chain Monte Carlo method is adopted in [3] to approximate $k$-means++ efficiently. In [2], a fast seeding approach is proposed, where a provably good clustering can be achieved even without any assumption on the dataset. More improvements can be found here [49, 28].

As can be seen from the afore-said related research, these two algorithms are only loosely related. In order to explore the relationship between kernel $k$-means and spectral clustering, much effort has been devoted, which narrows the gap between traditional $k$-means and spectral clustering to some extent [10].

In this paper, a unified framework between the traditional $k$-means and ratio-cut is firstly revisited, and an efficient clustering algorithm is then proposed based on this framework. It is worthwhile to highlight the main contributions of this paper as follows:

- The computational and memory overhead of our method are both linear with respect to the number of samples, and are independent of the number of clusters to construct. These properties mean that it is easily scalable and applicable to large practical problems.

- The optimization algorithm employed guarantees that no empty cluster will occur, with any initialization. In particular, our algorithm tends to produce a balanced partition.

- Extensive experiments on several real-world benchmark and facial datasets validate the advantages of the proposed algorithm compared to the state-of-the-art clustering algorithms.

**Notations:** Throughout the paper, matrices are written in boldface uppercase letters, and the vectors are denoted in boldface lowercase letters. For matrix $\mathbf{M} = [\mathbf{m}_1, \cdots, \mathbf{m}_n]^T \in \mathbb{R}^{n \times c}$, $m_{ij}$, $\mathbf{m}_i$ and $Tr(\mathbf{M})$ are used to represent the $(i, j)$-th entry, transpose of the $i$-th row and trace of $\mathbf{M}$, respectively. The $L_2$ norm of the vector $\mathbf{v} \in \mathbb{R}^n$ is defined as $\|\mathbf{v}\|_2 = \sqrt{\sum_{i=1}^{n} v_i^2}$. A matrix is referred to cluster indicator matrix, if each row of it has one and only one element equal to $1$ which indicates the cluster membership and the rest elements are all $0$. The set of all cluster indicator matrices is denoted by $\Phi^{n \times c}$. $\mathbf{1}$ is the column vector of all ones. $\mathbf{x}_i^{(j)}$ denotes the $j$-th nearest neighbor of sample $\mathbf{x}_i$.

## 2 An efficient clustering method

For convenience, let $\mathbf{Y} = [\mathbf{y}_1, \cdots, \mathbf{y}_n]^T \in \Phi^{n \times c}$ be an indicator matrix, i.e., $y_{ij} = 1$ if $\mathbf{x}_i \in A_j$, $y_{ij} = 0$ otherwise. Then the problem of classical ratio-cut can be written in the matrix form as

$$\min_{\mathbf{Y} \in \Phi^{n \times c}} Tr\left(\left(\mathbf{Y}^T\mathbf{Y}\right)^{-\frac{1}{2}} \mathbf{Y}^T(\mathbf{\Delta} - \mathbf{W})\mathbf{Y} \left(\mathbf{Y}^T\mathbf{Y}\right)^{-\frac{1}{2}}\right). \tag{2}$$

Many studies [25, 51, 45] have shown that the performance of graph-based clustering models can be enhanced by employing the doubly stochastic matrix, i.e., $\sum_i w_{ij} = 1$, $\sum_j w_{ij} = 1$. Therefore, we only focus on the graph-based method using the doubly stochastic matrix. With the definition of doubly stochastic matrix, the objective function of ratio-cut (2) can be simplified as

$$\max_{\mathbf{Y} \in \Phi^{n \times c}} Tr\left(\left(\mathbf{Y}^T\mathbf{Y}\right)^{-\frac{1}{2}} \mathbf{Y}^T\mathbf{W}\mathbf{Y} \left(\mathbf{Y}^T\mathbf{Y}\right)^{-\frac{1}{2}}\right). \tag{3}$$

Now, we turn our attention to the traditional $k$-means. For convenience, two lemmas are given firstly.

**Lemma 1.** For any vector $\mathbf{p} \in \mathbb{R}^n$, $\mathbf{p} \geq 0$, $\mathbf{p}^T\mathbf{1} = 1$, the following equation holds [30].

$$\sum_{i=1}^{n} p_i \left(\mathbf{x}_i - \bar{\mathbf{x}}\right)^T \left(\mathbf{x}_i - \bar{\mathbf{x}}\right) = \frac{1}{2} \sum_{i=1}^{n} \sum_{j=1}^{n} p_i p_j \left(\mathbf{x}_i - \mathbf{x}_j\right)^T \left(\mathbf{x}_i - \mathbf{x}_j\right), \tag{4}$$

where $\bar{\mathbf{x}} = \sum_{i=1}^{n} p_i \mathbf{x}_i$.

**Lemma 2.** For any vector $\mathbf{v} \in \mathbb{R}^n$ and $\mathbf{Y} \in \Phi^{n \times c}$, we have

$$Tr\left(\left(\mathbf{Y}^T\mathbf{Y}\right)^{-\frac{1}{2}} \mathbf{Y}^T \left(\mathbf{v}\mathbf{1}^T + \mathbf{1}\mathbf{v}^T\right) \mathbf{Y} \left(\mathbf{Y}^T\mathbf{Y}\right)^{-\frac{1}{2}}\right) = 2\sum_{i=1}^{n} v_i. \tag{5}$$

The proofs of the two lemmas above are shown in the supplementary material.

According to Lemma 1, the objective function of $k$-means (1) can be rewritten into

$$\min_{\mathbf{A}_1, \mathbf{A}_2, \cdots, \mathbf{A}_c} \sum_{i=1}^{c} \frac{1}{|\mathbf{A}_i|} \sum_{\mathbf{x}_j, \mathbf{x}_l \in \mathbf{A}_i} \|\mathbf{x}_j - \mathbf{x}_l\|_2^2 \tag{6}$$

$$\Leftrightarrow \min_{\mathbf{Y} \in \Phi^{n \times c}} \left(diag\left(\mathbf{Y}^T\mathbf{Y}\right)^{-1}\right)^T diag\left(\mathbf{Y}^T\mathbf{D}\mathbf{Y}\right) \tag{7}$$

$$\Leftrightarrow \min_{\mathbf{Y} \in \Phi^{n \times c}} Tr\left(\left(\mathbf{Y}^T\mathbf{Y}\right)^{-\frac{1}{2}} \mathbf{Y}^T\mathbf{D}\mathbf{Y} \left(\mathbf{Y}^T\mathbf{Y}\right)^{-\frac{1}{2}}\right), \tag{8}$$

where $\mathbf{D} \in \mathbb{R}^{n \times n}$ denotes the squared Euclidean distance matrix, $d_{ij} = \|\mathbf{x}_i - \mathbf{x}_j\|_2^2$. And $diag(\mathbf{M})$ represents a column vector consisting of diagonal elements of $\mathbf{M}$. According to Lemma 2 and the fact of $\mathbf{D} = \mathbf{v}\mathbf{1}^T + \mathbf{1}\mathbf{v}^T - 2\mathbf{X}\mathbf{X}^T$, where $\mathbf{v} = [v_1, \cdots, v_n]^T$, $v_i = \|\mathbf{x}_i\|_2^2$, the problem (8) then can be formulated as follows

$$\max_{\mathbf{Y} \in \Phi^{n \times c}} Tr\left(\left(\mathbf{Y}^T\mathbf{Y}\right)^{-\frac{1}{2}} \mathbf{Y}^T\mathbf{X}\mathbf{X}^T\mathbf{Y} \left(\mathbf{Y}^T\mathbf{Y}\right)^{-\frac{1}{2}}\right). \tag{9}$$

Obviously, both of the two ($k$-means, ratio-cut) can be unified into the following framework.

$$\max_{\mathbf{Y} \in \Phi^{n \times c}} Tr\left(\left(\mathbf{Y}^T\mathbf{Y}\right)^{-\frac{1}{2}} \mathbf{Y}^T\mathbf{G}\mathbf{Y} \left(\mathbf{Y}^T\mathbf{Y}\right)^{-\frac{1}{2}}\right), \tag{10}$$

where $\mathbf{G} = \mathbf{X}\mathbf{X}^T$ for $k$-means, and $\mathbf{G} = \mathbf{W}$ for ratio-cut. It is worth mentioning that SC aims to **maximize the mean of similarities** between points in the same cluster, while $k$-means aims to **minimize the mean of distances** between points in the same cluster, as shown in Eq. (3) and Eq. (8).

## 2.1 Our model

From the unified view between ratio-cut and traditional $k$-means , we have the following observations: 1. Ratio-cut adopted doubly stochastic matrix shares the same framework as the traditional $k$-means. 2. The mere difference between the two is that a heat kernel is usually adopted to measure the similarity in ratio-cut, but $k$-means uses the inner product to measure the similarity between samples. 3. The computational overhead of $k$-means is linear with respect to the number of samples, which should be credited to the property of the square Euclidean distance described in Lemma 1.

Although the models employing $k$-nearest neighbors graph ($k$-NNG) often yield more superior experimental performance compared to other algorithms, the optimization algorithm of them often has a high time complexity. In this paper, we focus on the situation where the number of samples in each cluster is strictly equal, with this assumption, the problem (10) can be convert to

$$\max_{\mathbf{Y} \in \Phi^{n \times c}} Tr \left( \left( \mathbf{Y}^T \mathbf{Y} \right)^{-\frac{1}{2}} \mathbf{Y}^T \mathbf{G} \mathbf{Y} \left( \mathbf{Y}^T \mathbf{Y} \right)^{-\frac{1}{2}} \right) \quad s.t. \mathbf{Y}^T \mathbf{Y} = \bar{n} \mathbf{I} \quad (11)$$

$$\Leftrightarrow \max_{\mathbf{Y} \in \Phi^{n \times c}} Tr(\mathbf{Y}^T \mathbf{G} \mathbf{Y}) \quad s.t. \mathbf{Y}^T \mathbf{Y} = \bar{n} \mathbf{I}, \quad (12)$$

where $\bar{n}$ is a constant. Note that problem (12) is still difficult to solve, so the balance constraint is removed making the problem easy to solve, and a distance matrix $\mathcal{D} = (d_{ij})_{n \times n}$ is adopted to avoid the trivial solution. Therefore, our model can be simplified as follows

$$\min_{\mathbf{Y} \in \Phi^{n \times c}} Tr(\mathbf{Y}^T \mathcal{D} \mathbf{Y}). \quad (13)$$

with

$$d_{ij} = \begin{cases} \tilde{d}_{ij} & f(\mathbf{x}_i, \mathbf{x}_j) = 1 \\ \gamma & f(\mathbf{x}_i, \mathbf{x}_j) = 0 \end{cases} , \quad \tilde{d}_{ij} = \begin{cases} \|\mathbf{x}_i - \mathbf{x}_j\|_2^2 & f(\mathbf{x}_i, \mathbf{x}_j) = 1 \\ 0 & f(\mathbf{x}_i, \mathbf{x}_j) = 0 \end{cases} ,$$

$\gamma$ represents the maximum value of all the elements in $\tilde{\mathbf{D}}$, and $f(\mathbf{x}_i, \mathbf{x}_j) = 1$ if $\mathbf{x}_i$ is among the $k$-nearest neighbors of $\mathbf{x}_j$ and vice versa. It is worth noting that there will be a trivial solution, i.e., all samples are grouped into the same cluster, if a similarity matrix $\mathbf{W}$ is adopted in Eq. (13), i.e. $\max_{\mathbf{Y} \in \Phi^{n \times c}} Tr(\mathbf{Y}^T \mathbf{W} \mathbf{Y})$. It is not difficult to see that our model aims to minimize the sum of the distances between samples in the same cluster. We term our algorithm as **k-sums**.

**Graph construction.** If only the similarity between samples is provided by the graph, then $\tilde{d}_{ij} = -log(s_{ij})$ is recommended to compute the distance between samples, where $s_{ij}$ denotes the similarity between samples $\mathbf{x}_i$ and $\mathbf{x}_j$. If the graph is not available, it is recommended to use the approximate nearest neighbor algorithm to construct the distance matrix. In this paper, k-d tree and EFANNA[14] are used for synthetic and benchmark datasets, respectively.

**Balanced partition.** It is worth noting that even without the constraint, $\mathbf{Y}^T \mathbf{Y} = \bar{n} \mathbf{I}$, the model still tends to produce a balanced partition. Assuming that the distance between any two data points is approximately equal to $\alpha$, then we have the following equation.

$$\min_{\mathbf{Y} \in \Phi^{n \times c}} Tr(\mathbf{Y}^T \mathcal{D} \mathbf{Y}) \approx \min_{\mathbf{Y} \in \Phi^{n \times c}} Tr(\mathbf{Y}^T \alpha (\mathbf{1}\mathbf{1}^T - \mathbf{I}) \mathbf{Y}) \quad (14)$$

$$\Leftrightarrow \min_{\mathbf{Y} \in \Phi^{n \times c}} \alpha \mathbf{1}^T \mathbf{Y} \mathbf{Y}^T \mathbf{1} - \alpha Tr(\mathbf{Y}\mathbf{Y}^T) \quad (15)$$

$$\Leftrightarrow \min_{\mathbf{Y} \in \Phi^{n \times c}} \alpha (n_1^2 + \cdots + n_c^2 - n). \quad (16)$$

Obviously, $n_i = n/c$ is the optimal solution of the problem.

There are three advantages of our model. **Efficient:** The problem (13) can be solved by a standard coordinate descent algorithm, and the optimization algorithm will have a low time and space complexity owing to sparsity of $\mathcal{D}$ and $\mathbf{Y}$. In fact, both computational and memory overhead are $O(nk)$, which are independent of the number of clusters to construct. **Effective:** Benefiting from the use of $k$-nearest neighbors graph, $k$-sums can be adopted to identify clusters that are non-linearly separable in input space. **Robust:** A constant $\gamma$ is used to replace the squared Euclidean distance between $\mathbf{x}_i$ and $\mathbf{x}_j$ if $f(\mathbf{x}_i, \mathbf{x}_j) = 0$, which means that our algorithm is not sensitive to outliers. In addition, only the Euclidean distance is involved in our model, so it avoids the trouble of measuring the similarity between samples.

# 3 Optimization

A standard coordinate descent algorithm is employed to optimize problem (13). For example, to solve the 1-th row of $\mathbf{Y}$, we fixed the other rows as constants. Let $\mathbf{Y}_r^T$ be the matrix $\mathbf{Y}$ with the first row removed, $\mathcal{D}_r$ be the matrix $\mathcal{D}$ with the first row and column removed. With these notations, the problem (13) then becomes

$$\min_{\mathbf{y}_1} Tr\left(\mathbf{Y}^T \mathcal{D} \mathbf{Y}\right) \tag{17}$$

$$\Leftrightarrow \min_{\mathbf{y}_1} Tr\left(\begin{bmatrix} \mathbf{y}_1 & \mathbf{Y}_r \end{bmatrix} \begin{bmatrix} d_{1,1} & \mathbf{v}^T \\ \mathbf{v} & \mathcal{D}_r \end{bmatrix} \begin{bmatrix} \mathbf{y}_1^T \\ \mathbf{Y}_r^T \end{bmatrix}\right) \tag{18}$$

$$\Leftrightarrow \min_{\mathbf{y}_1} Tr\left(\mathbf{y}_1 d_{1,1} \mathbf{y}_1^T\right) + 2Tr(\mathbf{Y}_r \mathbf{v} \mathbf{y}_1^T) \tag{19}$$

$$\Leftrightarrow \min_{\mathbf{y}_1} Tr(\mathbf{Y}_r \mathbf{v} \mathbf{y}_1^T) \Leftrightarrow \min_{\mathbf{y}_1} \mathbf{y}_1^T \tilde{\mathbf{Y}}^T \mathbf{d}_1, \tag{20}$$

where $\mathbf{v}^T = \begin{bmatrix} d_{1,2}, & d_{1,3}, & \cdots, & d_{1,n} \end{bmatrix}$, $d_{1,1} = 0$, $\tilde{\mathbf{Y}}$ represents the indicator matrix before $\mathbf{y}_1$ is updated. The update rules for other rows of $\mathbf{Y}$ can be similarly derived. Therefore, the optimal solution of $\mathbf{y}_i$ can be then obtained by

$$y_{il} = \begin{cases} 1 & l = \arg\min_j (\tilde{\mathbf{Y}}^T \mathbf{d}_i)_j \\ 0 & \text{otherwise.} \end{cases}, \tag{21}$$

Despite problem (21) is very simple, $O(kc)$ time is needed to obtain the optimal solution, which is very time-consuming, **when the number of clusters to construct is large.**

## 3.1 Acceleration

The meaning of the symbols involved in this subsection is given first. For any vector $\mathbf{v} \in \mathbb{R}^n$, $v_i$ and $v[i]$ are both used to represent the $i$-th element of $\mathbf{v}$, for convenience. Let $\mathbf{n} \in \mathbb{R}^c$ be the number of samples in each cluster. $\mathbf{p}, \mathbf{q}$ are vectors obtained by sorting $\mathbf{n}$, that is to say, $n[p_i] = q_i$ and $q_i$ is the $i$-th smallest element in $\mathbf{n}$.

Let $\mathbf{t}$ be the product of $\tilde{\mathbf{Y}}^T$ and $\mathbf{d}_i$, i.e. $\mathbf{t} = \tilde{\mathbf{Y}}^T \mathbf{d}_i$. That is to say, $t_j$ represents the sum of the elements of $\mathbf{d}_i$ belonging to the $j$-th cluster. There are at most $k$ elements in $\mathbf{t}$, called $\mathbf{t}^{(1)}$, which consists of values of $\mathbf{d}_i$ that are smaller than $\gamma$, and the remaining elements, called $\mathbf{t}^{(2)}$, are all integer multiples of $\gamma$. Next, we first find the minimum of $\mathbf{t}^{(1)}$ and $\mathbf{t}^{(2)}$, then compare the two to get the minimum value in $\mathbf{t}$. Obviously, $O(k)$ time is needed to find the minimum value in $\mathbf{t}^{(1)}$. What we care about is the time complexity of finding the minimum value in $\mathbf{t}^{(2)}$.

The cluster with the smallest number of samples in $\mathbf{t}^{(2)}$ corresponds to its minimum value, since the elements in $\mathbf{t}^{(2)}$ are all integer multiples of $\gamma$. In this way, we can easily find the minimum value of $\mathbf{t}^{(2)}$, but we have an easier way to implement it.

Based on the following facts, we found that the minimum value of $\mathbf{t}$ can also be obtained by directly comparing $\gamma q_1$ and the minimum value of $\mathbf{t}^{(1)}$.

- The cluster $p_1$ is involved in $\mathbf{t}^{(1)}$ instead of $\mathbf{t}^{(2)}$. Then the minimum value of $\mathbf{t}$ is the minimum value of $\mathbf{t}^{(1)}$, and $\gamma q_1 < \min(\mathbf{t}^{(1)})$ does not hold. The above conclusion holds.

- The cluster $p_1$ is involved in $\mathbf{t}^{(2)}$ instead of $\mathbf{t}^{(1)}$. Then $\gamma q_1$ is the minimum value of $\mathbf{t}^{(2)}$. Therefore we can found the minimum value of $\mathbf{t}$ by comparing $\gamma q_1$ and $\min(\mathbf{t}^{(1)})$. The above conclusion holds.

Fortunately, with the help of Lemma 3, we can always find the cluster with the least number of samples in $O(1)$ time. The detailed algorithm to solve problem (21) is summarized in Algorithm 1.

**Lemma 3.** Let $\mathbf{n} = [n_1, \cdots, n_c]^T \in \mathbb{R}^c$ be an ordered sequence consisting of positive integers, where $0 \le n_i \le N$, where $N < +\infty$ is a positive integer. $\mathbf{p}, \mathbf{q}$ are vectors obtained by sorting $\mathbf{n}$, i.e., $n[p_i] = q_i$. Only $O(1)$ time is required to maintain its order if $n_i$ is increased (reduced) by 1.

The proof of this lemma is shown in the support material again.

---

**Algorithm 1:** An efficient program for solving problem (21).

---

**Data:** $\mathbf{D} \in \mathbb{R}^{n \times k}$, where $d_{ij}$ denotes the squared Euclidean distance between $\mathbf{x}_i$ and $\mathbf{x}_i^{(j)}$.

$\quad$ $\mathbf{C} \in \mathbb{R}^{n \times k}$, where $c_{ij}$ denotes the cluster to which $\mathbf{x}_i^{(j)}$ currently belongs. $\mathbf{n}, \mathbf{p}, \mathbf{q} \in \mathbb{R}^c$,

$\quad$ Auxiliary vectors $\mathbf{u}, \mathbf{t}, \mathbf{v} \in \mathbb{R}^c$. the parameter $\gamma$.

**Result:** The position of the minimum value of $\mathbf{t} = \tilde{\mathbf{Y}}^T \mathbf{d}_i$, $l$.

**for** $j = 1, \cdots, k$ **do** { $t[c_{ij}] = 0$; $u[c_{ij}] = 0$; $v[c_{ij}] = 0$; }

**for** $j = 1, \cdots, k$ **do** { $t[c_{ij}] + = d_{ij}$; $u[c_{ij}] + = 1$; }

**for** $j = 1, \cdots, k$ **do**

$\quad$ **if** $v[c_{ij}] == 0$ **then**

$\quad\quad$ $v[c_{ij}] = 1$;

$\quad\quad$ $t[c_{ij}] + = \gamma(n[c_{ij}] - u[c_{ij}])$;

Find index $l$ such that $t_l$ is the smallest in set $\{t[c_{i1}], \cdots, t[c_{ik}]\}$, i.e. $\mathbf{t}^{(1)}$;

Modify the value of $l$ to $p_1$ if $\gamma q_1 < t_l$;

---

## 3.2 Time and space complexity

(1) The memory space is mainly occupied by the variables $\mathbf{D} \in \mathbb{R}^{n \times k}$, $\mathbf{C} \in \mathbb{R}^{n \times k}$, and vectors of length $c$, if the problem (21) is optimized with our algorithm. And $c$ is always less than $n$, so its space complexity is $O(nk)$. (2) $O(n)$ time is needed to initialize the vectors, $\mathbf{v}$, $\mathbf{u}$, $\mathbf{y}$ and $\mathbf{n}$. It takes $O(n)$ time to sort $\mathbf{n}$ using bucket sort algorithm. Note that once $\mathbf{n}$ is ordered, only $O(1)$ time is needed to hold its order in accordance with Lemma 2. $O(k)$ time is needed to update the label of the sample $\mathbf{x}_i$, seen from Algorithm 1. Once more, as $c$ is always smaller than $n$, therefore the computational complexity of our algorithm is $O(nk)$.

# 4 Experiments

In this section, we are going to show the performance of the proposed clustering method, $k$-sums, on six synthetic data sets and twenty benchmark data sets (12 middle-scale and 8 large-scale face datasets). The rest of this section is organized as follows: The description of the data sets are firstly represented, and then the comparison algorithms are introduced secondly and finally, clustering performance of all algorithms on both synthetic and real world data sets is shown.

## 4.1 Data sets

We conduct experiments on 6 synthetic data sets: Grid-10w, Grid-20w, Grid-40w, Grid-80w, Grid-100w, Outlier. The first 5 data sets are randomly generated multi-cluster data. In these data sets, there are 10k, 20k, 40k, 80k, and 100k clusters distributed in a spherical way, respectively, and each cluster contains 10 samples. We compare the running time of $k$-sums and $k$-means on the first 5 data sets to verify the efficiency of $k$-sums. The last toy dataset, Outlier, is designed to verify the robustness of $k$-sums against outliers.

The effectiveness of $k$-sums is mainly supported by the following benchmark datasets: Face-94, Face-95, Grimace, MPEG-7, JAFFE[26], FERET[35], FEI, Finger, GTdb, IMM, Olivetti[11] and Palm. The first three are widely used for face recognition, whose introduction and detailed statistics can be found here [2]. MPEG-7 is a widely used data set for shape matching evaluation consisting of 70 shape classes, each of whom is represented by 20 different images with high intra-class variability. In order to promote the research of individual facial expressions detection and facial recognition system evaluation, JAFFE containing 213 images of 7 facial expressions and FERET containing 11,338 samples of 994 subjects have been constructed, respectively. EFI, Finger, GTdb, and IMM are high-dimensional data sets with 307200, 65536, 21600, and 307200 respectively. Palm is a dataset used for gender recognition consisting of 2,000 hand images of 100 subjects. The detailed statistics of these datasets are summarized in Tab.5 shown in support material, and the 8 large-scale datasets for face recognition will be introduced in the next section.

Table 1: Performance of $k$-sums and $k$-means (Time: $k$-sums = k-d tree + Algo. 1).

| Time (s) | # Clusters | k-d tree | Algo. 1 | $k$-sums | $k$-means | Speed-up |
|---|---|---|---|---|---|---|
| Grid-10w | 10k | 03.90 | 0.17 | 4.07 | 56.28 | **13.85x** |
| Grid-20w | 20k | 11.89 | 0.39 | 12.28 | 218.70 | **17.81x** |
| Grid-40w | 40k | 39.89 | 0.85 | 40.74 | 491.37 | **12.06x** |
| Grid-80w | 80k | 143.47 | 2.01 | 145.48 | 2948.56 | **20.27x** |
| Grid-100w | 100k | 219.05 | 2.45 | 221.50 | 3416.70 | **15.43x** |

| | Precision | | Recall | | $F_1$ score | |
|---|---|---|---|---|---|---|
| Datasets | $k$-sums | $k$-means | $k$-sums | $k$-means | $k$-sums | $k$-means |
| Grid-10w | **0.979** | 0.760 | **0.980** | 0.854 | **0.980** | 0.804 |
| Grid-20w | **0.940** | 0.757 | **0.942** | 0.851 | **0.941** | 0.801 |
| Grid-40w | **0.940** | 0.738 | **0.941** | 0.837 | **0.940** | 0.784 |
| Grid-80w | **0.855** | 0.736 | **0.857** | 0.836 | **0.856** | 0.783 |
| Grid-100w | **0.887** | 0.702 | **0.889** | 0.809 | **0.888** | 0.752 |

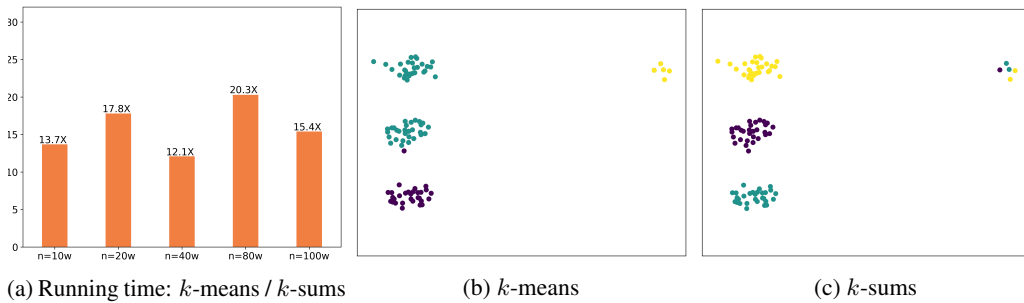

(a) Running time: $k$-means / $k$-sums          (b) $k$-means          (c) $k$-sums

Figure 1: Performance of $k$-sums $k$-means.

## 4.2 Experimental settings

We compared our method with several clustering algorithms, including traditional $k$-means [12], traditional Spectral Clustering (SC) [27], Scalable Spectral Clustering with cosine similarity (SSC) [6], Improved Anchor-based Graph Clustering based on multiplicative update optimization (AGC-I) [52], Fast Spectral Clustering with anchor graph for large hyperspectral images (FSC) [43], and Large scale Spectral Clustering via landmark-based sparse representation (LSC-R, LSC-K) [4].

To make a fair comparison with $k$-means implemented by scikit-learn [33], our code is implemented in C++. Both $k$-means and our code run on the Arch machine with 3.20 GHz i7-8700 CPU, 32 GB main memory. The parameters involved in these methods are described below.

In all anchor-based algorithms, the anchors are always generated in a random way if not explicitly stated in their papers, and the number of anchors $m$ is always set by $m = \min(n/2, 1024)$. In all graph-based algorithms, the method of constructing graphs follows the original paper, and the number of nearest neighbors $k$ is fixed at 20 for 6 synthetic and 12 middle-scale real world datasets. The $k$-means initialized in a random way is used, in these two-step algorithms. Every method takes 50 runs. The average results are reported.

## 4.3 Experimental results on synthetic data sets

It can be seen from Tab. 1 and Fig. 1(a) that our algorithm has a significant speed advantage over $k$-means and it is especially obvious when the number of samples is large. Specifically, over 15x and 13x speed-up can be obtained with respect to $k$-means on the synthetic dataset of 1 million and 1 hundred thousand samples, respectively. The speed-up of $dc/k$ can not be obtained, since the $k$-means is accelerated by triangle inequality [12] and the number of iterations of the two algorithms is different. As shown in Tab. 1 (part 2) and Fig. 1(b) and 1(c), $k$-sums shows good performance on both Grid-x and Outlier, which verifies the effectiveness and robustness of our model.

Table 2: Clustering accuracy on 12 real world datasets.

| Datasets | SC | $k$-means | LSC-R | LSC-K | SSC | AGCI | FSC | $k$-sums |
|---|---|---|---|---|---|---|---|---|
| Face-94 | 0.734 | 0.722 | 0.617 | 0.601 | 0.583 | 0.596 | 0.569 | **0.961** |
| Face-95 | 0.480 | 0.424 | 0.410 | 0.428 | 0.300 | 0.435 | 0.446 | **0.471** |
| FEI | 0.509 | 0.446 | 0.438 | 0.437 | 0.026 | 0.462 | 0.465 | **0.557** |
| FERET | 0.293 | 0.267 | 0.256 | 0.249 | 0.163 | 0.256 | 0.259 | **0.321** |
| Finger | 0.495 | 0.311 | **0.402** | 0.334 | 0.130 | 0.371 | 0.332 | 0.395 |
| Grimace | 0.785 | 0.733 | 0.666 | 0.705 | 0.741 | 0.654 | 0.655 | **0.983** |
| GTdb | 0.536 | 0.447 | 0.463 | 0.447 | 0.134 | 0.456 | 0.466 | **0.543** |
| IMM | 0.535 | 0.444 | 0.430 | 0.460 | 0.106 | 0.494 | 0.487 | **0.553** |
| JAFFE | 0.812 | 0.714 | 0.658 | 0.647 | 0.262 | 0.724 | 0.689 | **0.888** |
| MPEG-7 | 0.511 | 0.475 | 0.482 | 0.454 | 0.316 | 0.453 | 0.452 | **0.554** |
| Olivetti | 0.703 | 0.462 | 0.538 | 0.569 | 0.172 | 0.581 | **0.612** | 0.367 |
| Palm | 0.727 | 0.694 | 0.628 | 0.618 | 0.589 | 0.595 | 0.579 | **0.796** |

## 4.4 Experimental results on benchmark data sets

As shown in Tab. 2, it is clearly seen that $k$-sums achieves the highest performance in most cases comparing to the other clustering algorithms, showing the effectiveness of clustering based on $k$-nearest neighbors graph. Specifically, $k$-sums exceeds the second-best results 22.7%, 19.8%, 6.9%, 7.6% and 4.3% on Face-94, Grimace, Palm, JAFFE, and MPEG-7 datasets respectively, measured by clustering accuracy (ACC), and similar results can be obtained in terms of ARI and NMI shown in supplementary material. More importantly, $k$-sums has a lower computational complexity than traditional $k$-means, which has been verified theoretically and experimentally. However, it is meaningless to compare the running time on these middle-scale data sets, as the running time is not dominated by the number of samples on small-scale datasets.

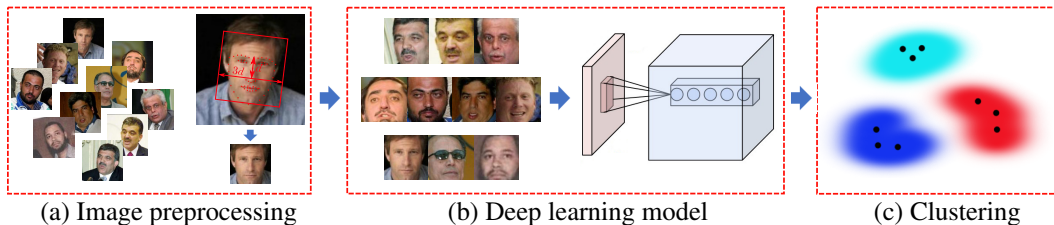

(a) Image preprocessing      (b) Deep learning model      (c) Clustering

Figure 2: Flow chart of deep clustering of unconstrained faces

## 5 Application: Face clustering

Clustering unconstrained faces is a important research topic in computer vision research and many algorithms have been proposed [40, 39, 46]. Among them, the two-stage framework is attracting more and more attention owing to its simplicity and efficacy [22, 32, 37]. Therefore, we employ it to verify the performance of the algorithm, i.e. 1. A deep model is adopted to map the face images into feature vectors. 2. The clustering model is performed in the feature space instead of the face images.

The experiment is conducted on 8 unconstrained facial datasets. The description of them as follows. WebFace [50] and CelebA [23] are two large-scale public datasets available for face recognition and verification problems. CALFW [54] and CPLFW [53] are two variants of LFW aiming at cross-age and cross-pose face recognition, respectively. CACD [5], Adience [15], and FERET [35] are constructed for cross-age face retrieval, age and gender recognition, and facial recognition system evaluation. A detailed description of them is given in the support material.

As shown in Fig. 2 The preprocessing process of data is as follows: Firstly, the pre-trained facial landmark detector in Dlib, is used to estimate the location of 68 facial landmarks. Then the images are rotated so that 20-th and 25-th facial landmarks are located on the same horizontal line, and cut out a square with a length of $3d$ centered on 31-th landmark ($d$ is the vertical distance from points 31 to 25). A trained deep model [47] is then employed to extract the features of the image. The $k$-nearest

Table 3: Running time (s) of $k$-sums and $k$-means ($k$-sums = EFANNA + Algo. 1).

| Datasets | # Samples | EFANNA | Algo. 1 | $k$-sums | $k$-means | Speed-up |
|---|---|---|---|---|---|---|
| Adience | 19,370 | 2.60 | 0.25 | 2.85 | 33.2 | **11.65x** |
| CPLFW | 11,652 | 1.60 | 5.23 | 6.83 | 28.4 | **4.158x** |
| FERET | 11,338 | 1.50 | 0.15 | 1.65 | 7.10 | **4.303x** |
| CACD | 163,446 | 17.9 | 2.70 | 20.6 | 2.7E3 | **131.1x** |
| CALFW | 12,174 | 1.50 | 7.90 | 9.40 | 27.5 | **2.926x** |
| WebFace | 494,414 | 93.7 | 16.7 | 1.1E2 | 5.7E2 | **5.182x** |
| PEAL | 30,863 | 3.30 | 0.65 | 3.95 | 19.2 | **4.861x** |
| CelebA | 202,599 | 26.9 | 4.30 | 31.2 | 2.3E2 | **7.372x** |

Table 4: Clustering performance of $k$-sums and $k$-means.

| | Precision | | Recall | | $F_1$ score | |
|---|---|---|---|---|---|---|
| Datasets | $k$-sums | $k$-means | $k$-sums | $k$-means | $k$-sums | $k$-means |
| Adience | **0.807** | 0.791 | 0.443 | **0.502** | 0.572 | **0.614** |
| CPLFW | **0.404** | 0.398 | 0.404 | **0.594** | 0.404 | **0.477** |
| FERET | **0.662** | 0.623 | **0.641** | 0.618 | **0.651** | 0.620 |
| CACD | **0.809** | 0.772 | **0.821** | 0.810 | **0.815** | 0.789 |
| CALFW | **0.760** | 0.693 | 0.748 | **0.749** | **0.754** | 0.719 |
| WebFace | **0.644** | 0.546 | **0.572** | 0.547 | **0.606** | 0.546 |
| PEAL | **0.861** | 0.735 | **0.881** | 0.842 | **0.871** | 0.785 |
| CelebA | **0.445** | 0.276 | **0.486** | 0.469 | **0.464** | 0.347 |

neighbors graphs are generated by EFANNA [14] with $k = 100$ for all facial datasets. Finally, the proposed algorithm is performed on those graphs to verify its performance.

From the results shown in Tab. 3, and Tab. 4, it shows to us that $k$-sums gains better performance in most of the cases, compared to $k$-means. And over 5x and 7x speed-up is obtained with respect to $k$-means on the two large-scale datasets (WebFace and CelebA). One more time, the speed-up of $dc/k$ can not be obtained, since the $k$-means is accelerated by triangle inequality [12] and the number of iterations of the two algorithms is different.

## 6   Conclusions

In this paper, we proposed an efficient clustering algorithm based on the unified view of $k$-means and ratio-cut. Both the computational and memory overhead of our method are linear with respect to the number of samples. Beside the linear computational complexity, the proposed algorithms have many other good properties. 1) The time complexity is not dependent on the number of clusters to construct, indicating that it is easily scalable and applicable to large practical problems; 2) The optimization algorithm employed guarantees that no empty cluster will occur, with any initialization. Specifically, our algorithm tends to produce a balanced partition; 3) Extensive experiments have been conducted on 20 real-world benchmark data sets to demonstrate the effectiveness and efficiency of our model.

## Broader Impact

Not applicable.

## Acknowledgments and Disclosure of Funding

This work was supported in part by the National Key Research and Development Program of China under Grant 2018AAA0101902, in part by the National Natural Science Foundation of China under Grant 61936014, Grant 61772427 and Grant 61751202, and in part by the Fundamental Research Funds for the Central Universities under Grant G2019KY0501. There are no financial conflicts of interest to disclose.

## Footnotes

[2]https://cswww.essex.ac.uk/mv/allfaces/

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
