[Supplementary Material]

# Supplementary Material

## 1 Proof of lemmas

**Lemma 1.** For any $p \geq 0$, $p^T 1 = 1$, the following equation holds.

$$\sum_{i=1}^{n} p_i \left(x_i - \bar{x}\right)^T \left(x_i - \bar{x}\right) = \frac{1}{2} \sum_{i=1}^{n} \sum_{j=1}^{n} p_i p_j \left(x_i - x_j\right)^T \left(x_i - x_j\right), \tag{1}$$

where $\bar{x} = \sum_{i=1}^{n} p_i x_i$.

**Proof.** On the left-hand side of this equation we have

$$\begin{aligned}
&\sum_{i=1}^{n} p_i (x_i - \bar{x})^T (x_i - \bar{x}) \\
&= \sum_{i=1}^{n} (p_i x_i^T x_i - 2 p_i x_i^T \bar{x} + p_i \bar{x}^T \bar{x}) \\
&= \sum_{i=1}^{n} p_i x_i^T x_i - 2\bar{x}^T \sum_{i=1}^{n} p_i x_i + \bar{x}^T \bar{x} \sum_{i=1}^{n} p_i \\
&= \sum_{i=1}^{n} p_i x_i^T x_i - 2\bar{x}^T \bar{x} + \bar{x}^T \bar{x} \\
&= \sum_{i=1}^{n} p_i x_i^T x_i - \sum_{i=1}^{n} p_i x_i^T \sum_{j=1}^{n} p_j x_j.
\end{aligned} \tag{2}$$

And on the right-hand side of this equation we have

$$\begin{aligned}
&\frac{1}{2} \sum_{i=1}^{n} \sum_{j=1}^{n} p_i p_j (x_i - x_j)^T (x_i - x_j) \\
&= \frac{1}{2} \sum_{i=1}^{n} \sum_{j=1}^{n} (p_i p_j x_i^T x_i + p_i p_j x_j^T x_j - 2 p_i p_j x_i^T x_j) \\
&= \frac{1}{2} \sum_{i=1}^{n} p_i x_i^T x_i + \frac{1}{2} \sum_{j=1}^{n} p_j x_j^T x_j - \sum_{i=1}^{n} \sum_{j=1}^{n} p_i p_j x_i^T x_j \\
&= \sum_{i=1}^{n} p_i x_i^T x_i - \sum_{i=1}^{n} p_i x_i^T \sum_{j=1}^{n} p_j x_j.
\end{aligned} \tag{3}$$

Eq.(2) and Eq.(3) are exactly the same, so Lemma 1 holds. $\blacksquare$

7 **Lemma 2.** For any vector $v \in R^n$ and indicator matrix $Y \in \Phi^{n \times c}$, we have

$$Tr\left(\left(Y^T Y\right)^{-\frac{1}{2}} Y^T \left(v\mathbf{1}^T + \mathbf{1}v^T\right) Y \left(Y^T Y\right)^{-\frac{1}{2}}\right) = 2\sum_{i=1}^{n} v_i, \tag{4}$$

**Proof .**

$$Tr\left(\left(Y^T Y\right)^{-\frac{1}{2}} Y^T \left(v\mathbf{1}^T + \mathbf{1}v^T\right) Y \left(Y^T Y\right)^{-\frac{1}{2}}\right) \tag{5}$$

$$=Tr\left(v\mathbf{1}^T Y \left(Y^T Y\right)^{-1} Y^T + v^T Y \left(Y^T Y\right)^{-1} Y^T \mathbf{1}\right) \tag{6}$$

$$=Tr\left(v\left[\begin{array}{ccc} n_1 & \cdots & n_c \end{array}\right] \left[\begin{array}{ccc} 1/n_1 & & \\ & \ddots & \\ & & 1/n_c \end{array}\right] Y^T \right. \tag{7}$$

$$\left. +v^T Y \left[\begin{array}{ccc} 1/n_1 & & \\ & \ddots & \\ & & 1/n_c \end{array}\right] \left[\begin{array}{c} n_1 \\ \vdots \\ n_c \end{array}\right]\right) \tag{8}$$

$$=Tr(v\mathbf{1}^T Y^T + v^T Y\mathbf{1}) \tag{9}$$

$$=2Tr(v^T Y\mathbf{1}) \tag{10}$$

$$=2Tr(v^T \mathbf{1}) \tag{11}$$

$$=2\sum_{i=1}^{n} v_i \tag{12}$$

8 **Lemma 3.** Let $v = [v_1, \cdots, v_c]$ represent an ordered sequence composed of positive integers,
9 where $0 \le v_i \le N$. Only $O(1)$ time is needed to maintain its order if $v_i$ is increased (reduced) by 1.

10 **Proof.** First, we initialize the sequence $F = [F_1, \cdots, F_N]$ and $L = [L_1, \cdots, L_N]$, where $F_i$ $(L_i)$
11 represents the position where the number $i$ first (last) appears in $v$ and $F_i = -1$ $L_i = -1$ if $i$ does
12 not appear in $v$. It takes $O(N)$ time to initialize $F$ and $L$. If $\tilde{v}_i \leftarrow v_i + 1$, we just need to swap $\tilde{v}_i$ and
13 $v_j, j = L_{v_i}$. Let $F_{\tilde{v}_i} = L_{v_i}, L_{\tilde{v}_i} = L_{v_i}$ if $F_{\tilde{v}_i} = -1$, $F_{\tilde{v}_i} \leftarrow F_{\tilde{v}_i} - 1$ otherwise, and $L_{v_i} \leftarrow L_{v_i} - 1$
14 if $L_{v_i} > F_{v_i}$, $L_{v_i} = -1$, $F_{v_i} = -1$ otherwise.

15 Example. $v = [1, 2, 2, 3, 5, 6, 6, 6, 7], 0 \le v_i \le 10$.

16 Initialize $F = [1, 2, 4, -1, 5, 6, 9, -1, -1, -1]$ and $L = [1, 3, 4, -1, 5, 8, 9, -1, -1, -1]$.

17 If $v_7 \leftarrow v_7 + 1(i = 7, v_i = 6, \tilde{v}_i = 7, L_{v_i} = 8, F_{v_i} = 6, L_{\tilde{v}_i} = 9, F_{\tilde{v}_i} = 9)$:

18 (1) Swap $6 + 1$ and $v_j, j = 8$, we have $v = [1, 2, 2, 3, 5, 6, 6, 6 + 1(\tilde{v}_i), 7]$

19 (2) Let $F_{\tilde{v}_i} \leftarrow F_{\tilde{v}_i} - 1$, we have $F[7] = 8$

20 (3) Let $L_{v_i} \leftarrow L_{v_i} - 1$, we have $L[6] = 7$

21 Finally, we have $v = [1, 2, 2, 3, 5, 6, 6, 7, 7]$, $F = [1, 2, 4, -1, 5, 6, 8, -1, -1, -1]$, and $L =$
22 $[1, 3, 4, -1, 5, 7, 9, -1, -1, -1]$. $F_i(L_i)$ still denotes the position where the number $i$ first (last)
23 appears in $v$.

24 If $v_4 \leftarrow v_4 + 1(i = 4, v_i = 3, \tilde{v}_i = 4, F_{v_i} = 4, L_{v_i} = 4, F_{\tilde{v}_i} = -1, L_{\tilde{v}_i} = -1)$:

25 (1) Swap $3 + 1$ and $v_j, j = 4$, we have $v = [1, 2, 2, 3 + 1(\tilde{v}_i), 5, 6, 6, 7, 7]$

26 (2) Let $F_{\tilde{v}_i} = L_{v_i}, L_{\tilde{v}_i} = L_{v_i}$, we have $F[4] = 4, L[4] = 4$

27 (3) Let $L_{v_i} = -1, F_{v_i} = -1$, we have $F[3] = -1, L[3] = -1$

28 Finally, we have $v = [1, 2, 2, 4, 5, 6, 6, 7, 7]$, $F = [1, 2, -1, 4, 5, 6, 8, -1, -1, -1]$, and $L =$
29 $[1, 3, -1, 4, 5, 7, 9, -1, -1, -1]$. $F_i(L_i)$ still denotes the position where the number $i$ first (last)
30 appears in $v$.

31 Note that the initialization of $F$ and $L$ only needs to be performed once at the beginning of the
32 program. It only takes $O(1)$ time to complete steps (1), (2) and (3).

# 2 Datasets used in Section 4

## 2.1 Description of datasets

Table 5: Description of datasets

|  | Datasets | # Samples | # Dimensions | # Subjects |
|---|---|---|---|---|
|  | Face-94 | 2,640 | 256 | 132 |
|  | Face-95 | 1,440 | 256 | 72 |
|  | FEI | 700 | 307,200 | 50 |
|  | FERET | 1,400 | 6,400 | 200 |
|  | Finger | 168 | 65,536 | 21 |
|  | Grimace | 360 | 256 | 18 |
| Middle-scale | GTdb | 750 | 21,600 | 50 |
|  | IMM | 240 | 307,200 | 40 |
|  | JAFFE | 200 | 65,536 | 10 |
|  | MPEG-7 | 1,400 | 6,000 | 70 |
|  | Olivetti | 900 | 2,500 | 10 |
|  | Palm | 2,000 | 256 | 100 |

## 2.2 Synthetic datasets

Table 6: Clustering results of $k$-sums and $k$-means on Grid-x

(a) The average p

|  | Datasets | k-d tree | Algo. 1 | $k$-sums | $k$-means | Speed-up |
|---|---|---|---|---|---|---|
|  | Grid-10w | 3.90 | 0.17 | 4.07 | 56.28 | **13.85x** |
| Running | Grid-20w | 11.89 | 0.39 | 12.28 | 218.70 | **17.81x** |
| Time (s) | Grid-40w | 39.89 | 0.85 | 40.74 | 491.37 | **12.06x** |
|  | Grid-80w | 143.47 | 2.01 | 145.48 | 2948.56 | **20.27x** |
|  | Grid-100w | 219.05 | 2.45 | 221.50 | 3416.70 | **15.43x** |

(b) The average performance of $k$-sums and $k$-means on synthetic datasets

|  | Precision | | Recall | | $F_1$ score | |
|---|---|---|---|---|---|---|
| Datasets | $k$-sums | $k$-means | $k$-sums | $k$-means | $k$-sums | $k$-means |
| Grid-10w | **0.979** | 0.760 | **0.980** | 0.854 | **0.980** | 0.804 |
| Grid-20w | **0.940** | 0.757 | **0.942** | 0.851 | **0.941** | 0.801 |
| Grid-40w | **0.940** | 0.738 | **0.941** | 0.837 | **0.940** | 0.784 |
| Grid-80w | **0.855** | 0.736 | **0.857** | 0.836 | **0.856** | 0.783 |
| Grid-100w | **0.887** | 0.702 | **0.889** | 0.809 | **0.888** | 0.752 |

(c) The standard deviation of $k$-sums and $k$-means on synthetic datasets

|  | Precision | | Recall | | $F_1$ score | |
|---|---|---|---|---|---|---|
| Datasets | $k$-sums | $k$-means | $k$-sums | $k$-means | $k$-sums | $k$-means |
| Grid-10w | **1.31E-03** | 7.77E-02 | **1.30E-03** | 6.02E-02 | **1.30E-03** | 7.02E-02 |
| Grid-20w | **9.09E-04** | 8.14E-02 | **8.71E-04** | 6.25E-02 | **8.89E-04** | 7.32E-02 |
| Grid-40w | **3.99E-04** | 7.74E-02 | **3.54E-04** | 6.01E-02 | **3.76E-04** | 7.00E-02 |
| Grid-80w | **6.33E-04** | 7.74E-02 | **6.29E-04** | 6.01E-02 | **6.29E-04** | 7.00E-02 |
| Grid-100w | **4.96E-04** | 5.19E-02 | **4.87E-04** | 3.99E-02 | **4.91E-04** | 4.67E-02 |

 **2.3 Real world datasets**

Table 7: Adjusted rand index on 12 real world datasets.

| Datasets | SC | $k$-means | LSC-R | LSC-K | SSC | AGCI | FSC | $k$-sums |
|----------|------|---------|-------|-------|-------|-------|-------|--------|
| Face-94 | 0.679 | 0.709 | 0.216 | 0.202 | 0.262 | 0.192 | 0.127 | **0.952** |
| Face-95 | 0.327 | 0.280 | 0.228 | 0.253 | 0.125 | 0.261 | 0.256 | **0.341** |
| FEI | 0.382 | 0.320 | 0.310 | 0.313 | 0.092 | 0.348 | 0.340 | **0.429** |
| FERET | 0.109 | 0.080 | 0.071 | 0.071 | 0.003 | 0.073 | 0.074 | **0.131** |
| Finger | 0.320 | 0.117 | **0.201** | 0.138 | 0.045 | 0.177 | 0.150 | 0.188 |
| Grimace | 0.777 | 0.716 | 0.547 | 0.594 | 0.636 | 0.549 | 0.500 | **0.970** |
| GTdb | 0.382 | 0.294 | 0.284 | 0.248 | 0.017 | 0.307 | 0.261 | **0.398** |
| IMM | 0.358 | 0.279 | 0.232 | 0.256 | 0.170 | 0.301 | 0.287 | **0.367** |
| JAFFE | 0.738 | 0.629 | 0.541 | 0.520 | 0.052 | 0.646 | 0.589 | **0.828** |
| MPEG-7 | 0.364 | 0.301 | 0.310 | 0.226 | 0.127 | 0.264 | 0.243 | **0.432** |
| Olivetti | 0.586 | 0.303 | 0.379 | 0.421 | 0.020 | 0.418 | **0.466** | 0.203 |
| Palm | 0.691 | 0.648 | 0.386 | 0.354 | 0.466 | 0.270 | 0.214 | **0.776** |

Table 8: Normalized mutual information on 12 real world datasets

| Datasets | SC | $k$-means | LSC-R | LSC-K | SSC | AGCI | FSC | Ours |
|----------|------|---------|-------|-------|-------|-------|-------|--------|
| Face-94 | 0.944 | 0.941 | 0.872 | 0.865 | 0.825 | 0.870 | 0.841 | **0.986** |
| Face-95 | 0.710 | 0.683 | 0.660 | 0.675 | 0.549 | 0.683 | 0.687 | **0.708** |
| FEI | 0.728 | 0.695 | 0.692 | 0.692 | 0.094 | 0.718 | 0.718 | **0.735** |
| FERET | 0.700 | 0.673 | 0.665 | 0.664 | 0.528 | 0.667 | 0.665 | **0.715** |
| Finger | 0.651 | 0.504 | **0.579** | 0.520 | 0.179 | 0.567 | 0.572 | 0.554 |
| Grimace | 0.932 | 0.901 | 0.848 | 0.871 | 0.807 | 0.862 | 0.853 | **0.984** |
| GTdb | 0.706 | 0.665 | 0.672 | 0.656 | 0.340 | 0.683 | 0.677 | **0.713** |
| IMM | 0.756 | 0.715 | 0.685 | 0.704 | 0.271 | 0.730 | 0.724 | **0.753** |
| JAFFE | 0.857 | 0.805 | 0.747 | 0.741 | 0.222 | 0.809 | 0.802 | **0.894** |
| MPEG-7 | 0.719 | 0.701 | 0.702 | 0.684 | 0.548 | 0.683 | 0.688 | **0.736** |
| Olivetti | 0.733 | 0.476 | 0.573 | 0.600 | 0.156 | 0.629 | **0.652** | 0.395 |
| Palm | 0.925 | 0.905 | 0.873 | 0.868 | 0.811 | 0.860 | 0.845 | **0.934** |

As in the paper, except SC, the best results are shown in **bold**.

# 3 Datasets used in Section 5

## 3.1 Description and running time

Table 9: Description of Datasets

| Datasets | # Samples | # Dimensions | # Classes |
|---|---|---|---|
| Adience | 19,370 | 256 | 2,284 |
| CPLFW | 11,652 | 256 | 3,930 |
| FERET | 11,338 | 256 | 994 |
| CACD | 163,446 | 256 | 2,000 |
| CALFW | 12,174 | 256 | 4,025 |
| WebFace | 494,414 | 256 | 10,575 |
| PEAL | 30,863 | 256 | 1,040 |
| CelebA | 202,599 | 256 | 10,177 |

Table 10: Running time(s) of $k$-sums and $k$-means on 8 face datasets

| Datasets | EFANNA | Algo. 1 | $k$-sums | $k$-means | Speed-up |
|---|---|---|---|---|---|
| Adience | 2.60 | 0.25 | 2.85 | 33.2 | **11.65x** |
| CPLFW | 1.60 | 5.23 | 6.83 | 28.4 | **4.158x** |
| FERET | 1.50 | 0.15 | 1.65 | 7.10 | **4.303x** |
| CACD | 17.9 | 2.70 | 20.6 | 2.7E3 | **131.1x** |
| CALFW | 1.50 | 7.90 | 9.40 | 27.5 | **2.926x** |
| WebFace | 93.7 | 16.7 | 1.1E2 | 5.7E2 | **5.182x** |
| PEAL | 3.30 | 0.65 | 3.95 | 19.2 | **4.861x** |
| CelebA | 26.9 | 4.30 | 31.2 | 2.3E2 | **7.372x** |

 ## 3.2 Clustering performance

Table 11: Clustering performance of $k$-sums and $k$-means

| Datasets | Precision | | Recall | | $F_1$ score | |
|---|---|---|---|---|---|---|
| | $k$-sums | $k$-means | $k$-sums | $k$-means | $k$-sums | $k$-means |
| Adience | **0.807** | 0.791 | 0.443 | **0.502** | 0.572 | **0.614** |
| CPLFW | **0.404** | 0.398 | 0.404 | **0.594** | 0.404 | **0.477** |
| FERET | **0.662** | 0.623 | **0.641** | 0.618 | **0.651** | 0.620 |
| CACD | **0.809** | 0.772 | **0.821** | 0.810 | **0.815** | 0.789 |
| CALFW | **0.760** | 0.693 | 0.748 | **0.749** | **0.754** | 0.719 |
| WebFace | **0.644** | 0.546 | **0.572** | 0.547 | **0.606** | 0.546 |
| PEAL | **0.861** | 0.735 | **0.881** | 0.842 | **0.871** | 0.785 |
| CelebA | **0.445** | 0.276 | **0.486** | 0.469 | **0.464** | 0.347 |

Table 12: The standard deviation of $k$-sums and $k$-means on 8 face datasets

| Datasets | Precision | | Recall | | $F_1$ score | |
|---|---|---|---|---|---|---|
| | $k$-sums | $k$-means | $k$-sums | $k$-means | $k$-sums | $k$-means |
| Adience | **1.71E-03** | 2.71E-03 | **1.22E-03** | 4.31E-03 | **1.18E-03** | 3.13E-03 |
| CPLFW | 1.25E-03 | **9.64E-04** | 1.27E-03 | **8.85E-04** | 1.26E-03 | **8.98E-04** |
| FERET | **1.83E-03** | 3.98E-03 | **1.86E-03** | 4.28E-03 | **1.09E-03** | 3.79E-03 |
| CACD | **1.73E-03** | 9.64E-02 | **8.46E-04** | 5.74E-02 | **1.27E-03** | 8.01E-02 |
| CALFW | **4.19E-03** | 4.61E-03 | 4.55E-03 | **3.42E-03** | 4.35E-03 | **3.72E-03** |
| WebFace | **7.74E-04** | 9.56E-02 | **3.24E-04** | 1.03E-01 | **3.60E-04** | 9.93E-02 |
| PEAL | **3.41E-03** | 4.90E-03 | 2.57E-03 | **2.50E-03** | **2.97E-03** | 3.47E-03 |
| CelebA | **5.77E-04** | 4.18E-02 | **5.18E-04** | 1.74E-02 | **5.07E-04** | 3.80E-02 |