[Reviews · NeurIPS 2020]

Review 1

Summary and Contributions: The paper introduces a computationally efficient clustering algorithm whose objective is the minimisation of the sum-of-square (SS) distances between points in the same cluster. To allow for more flexible clustering, only the k-nearest neighbour distances of each point are considered in the objective. In addition a simple truncation of large distances is introduced to mitigate the effect of outliers. The method is shown to produce promising practical performance in comparison with existing techniques both computationally and in terms of clustering performance.

Strengths: The main strength of the paper lies in the practical aspects of the proposed algorithm, in that it is efficient and the performance is extremely compelling. Despite the combinatorial nature of the objective, the authors have devised an intelligent and efficient approach to update the cluster assignments iteratively with minimal computational cost.

Weaknesses: The paper is, however, not clearly written. There are numerous instances of notation which is either not defined or is used before being defined. The discussion relating to the connection between k-means and clustering with graph cuts is also not new, see for example [1]. Furthermore, while the authors claim that this is the basis for their algorithm, the justification for this motivation does not come through. The objective is clear enough without this motivation. [1] Dhillon, Inderjit, Yuqiang Guan, and Brian Kulis. "A unified view of kernel k-means, spectral clustering and graph partitioning." Dept. Comput. Sci., Univ. Texas at Austin, Austin, Tech. Rep. TR-04-25 (2005).

Correctness: As far as I can tell nothing in the paper is incorrect

Clarity: The paper is not clearly written, especially in the technical parts. This is a major weakness of the paper. I will give a few specific examples in a later section

Relation to Prior Work: The existing work on the relationship between kmeans and graph cuts is not sufficiently discussed

Reproducibility: Yes

Additional Feedback: EDIT: I am satisfied by the response of the reviewers that they will address the issues of clarity, after which I believe the paper represents a valuable contribution. I commend the authors for what appears to be an innovative algorithm with extremely good practical performance. I believe the paper could be a very influential one, but I feel the presentation of the work needs to be modified and improved. (1) Try to be clearer on the motivation for using the connection between kmeans and graph cuts. I think there are a few too many concessions which are made. For example, you begin with ratio cut, then change to normalised cut when you assert that the affinity matrix is made doubly stochastic. This seems only for the purpose of making the objective look like the objective you use, since in your algorithm you do not normalise the affinity matrix (which in your case is more like a distance matrix). I think that the reformulation of the sum of squared distances between points and their mean in terms of the sum of inter-point squared distances (Lemma 1... note also that this is a well known fact and doesn't need its own lemma), combined with the use of only nearest neighbours to avoid the limitations of kmeans in terms of cluster shape is sufficient to motivate your formulation. (2) The notation and general presentation of the technical material should be reviewed and improved. It currently makes the paper unnecessarily difficult to follow. Some steps in the derivations are also not explained. (3) Have the paper proof-read to remove grammatical errors. In addition, I have the following specific comments/queries 1. Line 30: the computational complexity of spectral clustering is (cn^2) since only c (the number of clusters) eigenvectors are needed, not all n 2. Line 59: typo: "is difficulty" 3. Line 63: The sentence is grammatically incorrect 4. Line 80: You mention that you use boldface for vectors, but this is not the case in the paper 5. Line 84: Introduce the dimension of the set of cluster indicator vectors \Phi at this point, i.e., introduce the indices n and c when you first introduce \Phi 6. Section 2: The discussion of graph cuts is incomplete. You introduce the concept of vertices and nodes without defining them, nor mentioning their equivalence 7. Eq 3: Define the complement of A, \bar A 8. Line 90: define the indicator function 9. Line 93: what was previously the degree matrix D is suddenly changed to \Delta 10. Lemma 1: typo, p should be greater than zero! also, define what it means for a vector to be greater than zero 11. Line 111: bistochastic should be doubly stochastic 12. Equation 12: You introduce the constraint of balanced cluster size, but then subsequently ignore it. It looks as though this is introduced just to get rid of some of the terms in the trace, so that the problem is easier to solve. That is not a problem in and of itself, we often make concessions to get approximate solutions to a problem, but you should not enforce it as a constraint unless you actually use it that way. 13. The description of the matrix G is very poor and requires considerably more clarity. 14. Equation 14: You introduce the matrix U_{(i)} without defining it. As far as I can tell this is a permutation matrix to swap the first and i-th rows/columns, but this should be made precise and explicit. 15. Line 138: what is mean by "Ỹ represents the probability indicator matrix"? 16. Line 141: There is some stray bold font in this line. Where is that from? 17. Figures 1 (b) and (c) are not explained. I presume these are clustering solutions from a data set with outliers, but it isn't clear. 18. In the tables you have ignored when SC performs best. Why is this the case? Also, mention that bold face in the tables indicates the best performance. 19. In the experiments you refer to both accuracy and precision. Do you mean these to be the same? 20. Line 230: typo: "tow" should be "two". Finally, I think that the practical performance of the algorithm is almost sufficient to warrant publication. However, there are currently just too many minor errors and imprecisions in the paper. If the authors can fix these, then I would be encouraged to raise my score.


Review 2

Summary and Contributions: The paper suggests a new algorithm for the k-means problem. The algorithm is the coordinate descent algorithm on a new optimization problem defined in the paper. This optimization emerges from the phrasing of k-means and ratio cuts in the language of linear algebra. The authors run experiments examining the new algorithm. The datasets are (a) synthetic and (b) face images. They compared the new algorithm to several baselines like k-means++ and spectral clustering.

Strengths: The k-means problem is a fundamental problem in machine learning, and fast algorithms are desired. Even though the problem is 50 years old and researched thoroughly, the authors bring their new viewpoint and suggest a new algorithm.

Weaknesses: * There are several assumptions made during the text, for example, the distance measure is doubly stochastic, the clusters are of similar size. These assumptions might not always hold. * It's unclear what is the novelty of the work. The optimization problem presented in the paper is not stated as one of the contributions of the paper. So I deduce that the algorithm is the novel part, but it is an efficient standard optimization technique (coordinate descent). * The algorithm adds a new hyperparameter to the problem: how many nearest neighbors (k) to use.

Correctness: In the experience section 5, the accuracy of the algorithms is calculated using a reference clustering. What is this reference? Are the datasets labeled and the labels serve as a proxy of the optimal clustering?

Clarity: The paper is well-written. A few suggestions to improve it: - Give more intuition for the different equations on page 4 - State all the assumptions in one place, at the beginning - D tilde on line 122 is unnecessarily

Relation to Prior Work: It wasn't clear if the optimization problem in Equation (13) was known in the literature. It will be great if a subsection was added on the literature of k-means from the linear algebra point of view.

Reproducibility: Yes

Additional Feedback: Thanks for your feedback. I'm interested in better understanding your proposed model's novelty compared to the known literature (for example, the one suggested by Reviewer #1). I am looking forward reading your revised paper.


Review 3

Summary and Contributions: The work proposed a clustering method (called "k-sums") based on "a unified view of k-means and ratio-cut" algorithms. A number of benefits were demonstrated with the method, including a time complexity that is independent of cluster numbers, directly obtaining the clustering results without post-processing, and improved empirical results on the experimental datasets. ================= Although concerns still exist, the overall score was increased to "weak acceptance" in response to other reviewers' rough consensus.

Strengths: 1. Clustering is a fundamental and important problem in machine learning and data analysis. The proposed work fits the scope of NeurIPS community quite well. 2. The proposed clustering algorithm runs much faster than the classical k-means algorithm, which is a good characteristic that is often highly desired in practical applications.

Weaknesses: 1. On novelty: The relationship between k-means type algorithms and the graph-cut based algorithms was well studied in machine learning literature, for example, Dhillon et al. 2004. Deriving algorithms from this viewpoint is not surprising, and the novelty of the proposed work seems might be challenged. 2. On evaluation: The evaluation needs to be improved. The proposed clustering method uses k-d tree to construct a k-nearest neighbors graph as an intermediate step. Correspondingly, it is suggested to include the clustering performance of k-d tree as a baseline to compare. 3. With the constructed k-nearest neighbor graph, why not include the performance of ratio-cut or normalized-cut algorithms as another comparison baseline? This will help figure out where the improvement of the proposed approach come from.

Correctness: I didn't verify all the mathematical details, while the result is not surprising to me.

Clarity: Although the writing is roughly acceptable, there is much room for improvement. A thorough proofreading and re-organization are strongly suggested. Line 99, "For any p<=0", should it be "p>=0"? Line 156, "It takes O(n + c) time to sort v using bucket sort algorithm". Why?

Relation to Prior Work: Roughly acceptable, but can be improved significantly.

Reproducibility: Yes

Additional Feedback: Ref. "Weaknesses" section.


Review 4

Summary and Contributions: I read the rebuttal and overall am satisfied with the response, and thus would keep my score. A review of the unified framework of kmeans and ratio-cut is introduced in this paper. And then, a novel and efficient clustering algorithm is proposed based on this framework.

Strengths: The most attractive contribution of this article is the performance in terms of computational overhead and effectiveness. Specifically, (1) the model proposed in this article called k-sums has a computational complexity of O(nk) that is independent of the number of clusters to construct, which is a very surprising and important result; (2) the effectiveness of k-sums has been verified through experiments conducted on 20 benchmark datasets ranging between a few hundred and nearly .5 million samples and 5 synthetic datasets ranging between 100k and 1 million samples. Pros: 1, To be honest, clustering methods with excellent performance are not rare in the literature. However, K-means is still the clustering algorithm that engineers prefer. This is mainly because those algorithms often have a high time complexity or involve nuisance hyper-parameter(s). Only one parameter is involved in k-sums, i.e., the number of nearest neighbors that is an integer and is easy to tune, which is very important in practical applications. 2, The method used to optimize the objective function of k-sums is a standard coordinate descent algorithm. However, the idea of ​​speeding up the optimization process is very interesting. Specifically, it makes the computational overhead is independent of the number of clusters to construct. This conclusion is very rare in partition-based clustering methods. 3, Extensive experiments on 12 middle-scale and 8 large-scale benchmark datasets validate the effectiveness and efficiency of the proposed algorithms compared to the state-of-the-art clustering algorithms.

Weaknesses: 1,The equivalent symbol (\LeftRightArrow) between formulas 12 and 13 makes me very confused. It is recommended that the author use a clearer way to express the derivation of the model. 2, The review of k-means and spectral clustering takes up too much space. It is recommended to delete some content of this section. 3, This submission has few typos as follows. (1) In page 7, line 202, "18x and 29x" should be "15x and 13x", as shown in supplementary material. (2) In page 5, line 141, "ng, " should not be shown in bold. (3) Some equations are not followed by punctuation marks, such as (4), (5), (7), and (19), etc. 4, The sensitivity of the algorithm to the parameter K should be supplemented. If the performance of the algorithm is greatly affected by the value of K, then this will be a disadvantage of the algorithm. Overall, this paper presented an algorithm with excellent performance in terms of both computational overhead and effectiveness. Such an algorithm should be very popular in practical applications.

Correctness: yes

Clarity: yes

Relation to Prior Work: yes

Reproducibility: Yes

Additional Feedback: see weakness

[Author Response · NeurIPS 2020]

**Reviewer #1**

Thank you for your affirmation and encouragement. Your suggestions are of great help to the improvement of the paper.
**Issue 1:** While the authors claim that the connection between $k$-means and ratio-cut is the basis for their algorithm, the justification for this motivation does not come through. The objective is clear enough without this motivation.
**Reply 1:** As you said, the objective can indeed be clearly expressed without this motivation. However, our idea of using $k$-nearest neighbor graph and Euclidean distance metric in the objective function comes from the observation of the unified view. Therefore, we take it as the motivation of our model. I am sorry the paper is not clearly written.
**specific comments/queries: 9.** In order to distinguish it from the distance matrix involved in $k$-means, I expressed the degree matrix of the graph as $\mathbf{\Delta}$. I will also modify the degree matrix involved in Section 1 to $\mathbf{\Delta}$. **10.** Vector $\mathbf{p}(\mathbf{p} \geq 0, \mathbf{p}^T \mathbf{1} = 1)$, used to represent the weight of each item. **12.** As you said, my statement does have some improprieties. I will separate the model with constraint from the final model we employed so that readers can clearly understand the concessions made by the model. **15.** $\tilde{\mathbf{Y}}$ represents the indicator matrix $\mathbf{Y}$ but the $i$-th row of $\tilde{\mathbf{Y}}$ does not need to be optimized. **18.** As mentioned in Section 1, SC can usually obtain satisfactory performance with high complexity. Therefore most studies are devoted to reducing its complexity, resulting in the performance of the proposed algorithm is usually not as good as SC. So, in the tables, we have ignored the cases where SC performs best. **19.** Accuracy and Precision are two different metrics. The calculation of Accuracy needs the help of bestmap that is very time-consuming. Precision often appears together with recall, is a metric that is more suitable for large-scale data. The clustering performance in terms of precision, recall, and $F_1$ score is shown in the supplementary material.

In addition, I will seriously consider the suggestions in **1, 2, 3, 4, 5, 6, 7, 8, 11, 13, 14, 16, 17, and 20**, and carefully modify the submission, regardless of whether the submission is accepted or not.

**Reviewer #2**

**Issue 1:** There are several assumptions made during the text, for example, the distance measure is doubly stochastic, the clusters are of similar size. These assumptions might not always hold.
**Reply 1:** (1) Indeed, the matrix $\mathbf{W}$ cannot be guaranteed to always be a double random matrix. But all we need to know is that the equivalence between KM and Ratio-cut holds under some conditions. (2) As the reviewer said, this assumption (the clusters are of similar size) is a concession that makes the problem easy to solve.
**Issue 2:** It's unclear what is the novelty of the work. The optimization problem presented is not stated as a contribution.
**Reply 2:** The proposed model is indeed one of the important contributions of this article. As you said, this was ignored when listing contributions to this article. We will declare this contribution in the list in subsequent versions.
**Issue 3:** The algorithm adds a new hyper-parameter to the problem (i.e. the number of nearest nighbors).
**Reply 3:** You are right. Almost all graph-based methods involve this parameter, which is difficult to avoid. Fortunately, the performance of our model is not sensitive to this parameter. Satisfactory performance is achieved with K is fixed at 20, which is also mentioned in Section 4.2.
**Issue 4:** The accuracy of the algorithms is calculated using a reference clustering. What is this reference?
**Reply 4:** Our model was evaluated on datasets with labels, where the labels serve as a proxy of the optimal clustering.

**Reviewer #3**

First of all, thank you for your valuable comments. However, this paper does not actually have the problems described in suggestions 2 and 3. The main reason for this misunderstanding is that the paper is not clearly written. To this end, I will carefully revise my paper based on the reviewers' suggestions.
**Issue 1:** Deriving algorithms from the relationship between $k$-means and graph-cut is not surprising, and the novelty of the proposed work seems might be challenged.
**Reply 1:** As the reviewer #1 said, the relationship between $k$-means and ratio-cut is the basis of our algorithm. It is used to illustrate the motivation of the proposed model, and this relationship is not a contribution of this article.
**suggestion 2:** it is suggested to include the clustering performance of k-d tree as a baseline to compare since the proposed method uses k-d tree to construct a $k$-nearest neighbor graph as an intermediate step.
**Reply 2:** Sorry, I don't get the point of this suggestion. How do I compare the clustering algorithm with the k-d tree? The k-d tree seems to be an algorithm used to quickly find the neighbors of samples. I don't know how to get the cluster structure from this algorithm.
**suggestion 3:** Why not include the performance of ratio-cut or normalized-cut algorithms as a comparison baseline?
**Reply 3:** The problems of both ratio-cut and normalized-cut are NP-hard. The normalized spectral clustering (SC) considered in our experiments can be seen as a relaxed version of normalized-cut. And the unnormalized SC derived from ratio-cut have a similar performance with normalized SC.

**Reviewer # 4**

Thank you for your affirmation and encouragement. As you said, the most attractive contribution of this article is the performance in terms of computational overhead. To the best of our knowledge, the proposed method is the first graph-based algorithm with the computational complexity that is independent of the product of $n$ (the number of samples) and $c$ (the number of clusters). In addition, I will seriously consider your suggestions and revise this paper.

[Meta-Review · NeurIPS 2020]

The authors present an efficient algorithm for the sum-of-square objective. The proposed method has very impressive experimental performances and could be of interest for a broad audience. The paper contains a number of typos that should be fixed before publication.